# MAG Welding Process with Micro-Jet Cooling as the Effective Method for Manufacturing Joints for S700MC Steel

Tomasz Węgrzyn [1,*], Tadeusz Szymczak [2], Bożena Szczucka-Lasota [1] and Bogusław Łazarz [1]

[1] Faculty of Transport and Aviation Engineering, Silesian University of Technology, 40-119 Katowice, Poland; Bozena.Szczucka-Lasota@polsl.pl (B.S.-L.); Boguslaw.Lazarz@polsl.pl (B.Ł.)

[2] Motor Transport Institute, ITS Jagiellońska 80, 03-301 Warszawa, Poland; tadeusz.szymczak@its.waw.pl

[*] Correspondence: tomasz.wegrzyn@polsl.pl; Tel.: +48-504-816-362

**Abstract:** Advanced high-strength steel (AHSS) steels are relatively not very well weldable because of the dominant martensitic structure with coarse ferrite and bainite. The utmost difficulty in welding these steels is their tendency to crack both in the heat affected zone (HAZ) and in weld. The significant disadvantage is that the strength of the welded joint is much lower in comparison to base material. Adopting the new technology regarding micro-jet cooling (MJC) after welding with micro-jet cooling could be the way to steer the microstructure of weld metal deposit. Welding with micro-jet cooling might be treated as a very promising welding S700MC steel process. Tensile and fatigue tests were mainly carried out as the main destructive experiments for examining the weld. Also bending probes, metallographic structure analysis, and some non-destructive measurements were performed. The welds were created using innovative technology by MAG welding with micro-jet cooling. The paper aims to verify the fatigue and tensile properties of the thin-walled S700MC steel structure after welding with various parameters of micro-cooling. For the first time, micro-jet cooling was used to weld S700MC steel in order to check the proper mechanical properties of the joint. The main results are processed in the form of the Wöhler's S–N curves (alternating stress versus number cycles to failure).

**Keywords:** civil engineering; mechanical engineering; transport; smart city; micro-jet welding; mechanical resistance; microstructure; mini-specimen; fracture; fatigue limit

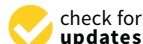

## 1. Introduction

A smart city is an urban area that uses different types of electronic methods and sensors to collect data [1]. This includes also data about transportation systems and communication technology [2–4]. Transport based on lighter and more durable means of transport will play an important role in the smart city. Authors put especial attention to the necessity of using modern materials and the development of various technologies (including welding) to create a Smart City model. In the modern automotive industry, high-strength martensitic steels from the AHSS group are increasingly used. Still welding of AHSS steels does not meet the satisfactory results because of the much lower tensile [5,6] and fatigue strength [6–8] of the joint compared to the base material. The differences of these properties are the consequence of chemical composition and structure of the base material and weld, which results in a need to use electrode wires with an increased nickel and molybdenum content and much lower sulfur content that affects the joint's structure and mechanical properties. The weld metal deposit contains mainly martensite and bainite with coarse ferrite while the base material contains mainly martensite and rather much fragmented ferrite. S700MC advanced high-strength steel (AHSS) can be obtained as hot-rolled, cold-rolled, hot-dip galvanized, and electro galvanized products. The material range includes thin sheet steel with a thickness range from 0.4 mm to 3 mm. This form of steel is used in slimming various structures (mobile platforms, containers, etc.,).

The increase in the use of modern AHSS steels in the automotive industry results from the possibility of reducing the thickness of car body sheets with a simultaneous improvement in the mechanical properties of the structure compared to the use of conventional low alloy steels. For construction of various modes of transport, the usage of high-strength martensitic steels from the group becomes increasingly popular [9–12]. These steel grades are relatively not very well weldable. That joint has good mechanical properties after welding, especially high impact toughness and elongation. The weld is not prone to cracks. The utmost difficulty in welding these steels is their tendency to crack in the weld and heat affected zone [8–10]. The main parameter determining the weldability of the steels is the carbon content and the carbon equivalent. In S700MC steels, the CET coefficient (carbon content equivalent, which, apart from C, also takes into account the impact on the weldability of steel of other elements: Mn, Mo, Cr, Cu, Ni) is similar to that of low-alloy steels [13].

This study analyzes the welding possibilities of AHSS steel with micro-jet cooling for the first time [7,14,15]. Welding high-strength steels is difficult. It is important to properly select the process and determine the correct parameters. The experimental results showed that the laser HSLA welds failed in a ductile necking/shear failure mode and the ductile failure was initiated at a distance away from the crack tip near the boundary of the base metal and heat affected zone [16]. Micro-jet cooling provides different cooling conditions than classic "macro" cooling. It is related to the so-called "scale effect," mentioned by other authors [17,18], who found, for example a great influence of micro-jet cooling on ferrite grinding during welding of low-alloy steel, which could not be obtained during "macro" cooling. After micro-jet cooling in the same welding conditions 70% of acicular ferrite was obtained, while after welding with classic "macro" cooling it was possible to obtain only 50% acicular ferrite, the most favorable phase guaranteeing high impact toughness at low temperatures and attractive ductility. For this reason, there was no attempt to deal with macro cooling in this research. The micro-jet cooling enables for selective and spot cooling of the welds, which allows controlling the microstructure of the welds. This method has proven successful in welding low-alloy steels and aluminum alloys [19].

Because of the welding with micro-jet cooling in welded joints made of low-alloy steels, high impact strength at low temperatures is obtained. In the case of aluminum alloy, higher strength and much better electrical conductivity (what was the purpose of this research) is reached as the effect of the joining process [18].

Fatigue and tensile tests are important destructive experiments in the quality assessment of AHSS welds [20–23]. Generally, during AHSS steels welding, a thermodynamic analysis of the process is required. In order to reduce welding stresses and to refine the ferrite, it is recommended to limit the linear energy during welding up to 4 kJ/cm [24–26]. In order to reduce the hydrogen content in the weld, it is recommended to use preheating and control the temperature of the interpass layers [27,28]. During welding, single hydrogen atoms H are formed in the weld, where they combine with the $H_2$ molecule (recombination effect). Accumulation of hydrogen inside the metal causes the formation of internal pressure, causing internal stresses of the material, which in turn lead to the formation of hydrogen-induced cracking (HIC) [29–34]. HIC cracking in S700MC steel is initiated mainly at the martensite-ferrite grain boundaries and in contact with ferrite [35–38]. In order to improve the process of increasing its repeatability, it was decided to weld S700MC steel with various parameters of micro-jet cooling. Micro-jet cooling consists in passing a narrow stream of gas through a specially designed injector, coupled to the welding head. In welding conditions, the following gases are used for this process: helium, argon, carbon dioxide, air, nitrogen, various mixtures. The gas pressure is in the range 0.4–0.7 MPa and the stream diameter is in the range of 50–80 μm.

Steel S700MC steel joints were not made using micro-jet cooling, so it is important to know the fatigue strength of this steel, because fatigue strength is one of the fundamental data for concluding on the quality of the joint [39–43]. This is an important reason for the research undertaken.

## 2. Materials and Methods

Welded (BW) butt joints of S700 MC steel with a thickness of 3 mm were made. The MAG (135) welding method in the flat position (PA) was applied according to the requirements of EN 15614-1 standard. Preparation of the material for a single-stitch welding and a finished 3-mm thick weld is shown in the Figure 1.

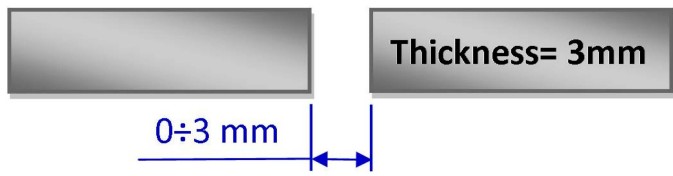

**Figure 1.** Preparation of the element for metal active gas (MAG) welding with micro-jet cooling.

It was decided to produce welds with the use of MAG (metal active gas) process testing the following gas mixture acting as a shielding gas mixture: Ar + 18% $CO_2$ (according to the PN-EN 14175 standard). Two different electrode wires (recommended for welding S700MC steel) were used for welding in order to more accurately investigate the effect of micro-jet cooling on the mechanical properties of a joint made in various welding conditions. The use of two different wires helps to judge that micro-jet cooling is beneficial when welding this steel. All of the samples were welded with two electrode wires (UNION X90 and UNION X96):

- EN ISO 16834-A G 89 6 M21 Mn4Ni2CrMo-UNION X90 (C 0.10, Si 0.8, Mn 1.8, Cr 0.35, Mo 0.6, Ni 2.3);
- EN ISO 16834-A G 89 5 M21 Mn4Ni2,5CrMo-UNION X96 (C 0.11, Si 0.78, Mn 1.9, P 0.01, S 0.009, Cr 0.35, Mo 0.57, Ni 2.23, V 0.004, Cu 0.02, Ti 0.057, Zr 0.001, Al 0.002).

Welded joints were made by MAG welding (135) using the Migatronic MIG-A Twist burner (Migatronic, Fjerritslev, Denmark) with intelligent arc control function. The burner is equipped with the IAC™ Intelligent Arc Control function, which provides a lower value of input energy, less distortion, while maintaining the mechanical properties of the material. The used technology ensures much lower power consumption compared to traditional welding machines [44]. Welding tests were performed in the gas shielding mixture M21 (82% Ar + 18% $CO_2$) with the assumed gas flow of 13 l/min. The input energy during the welding of thicker sheets (3 mm) was below the recommended value of 4 kJ/cm. All welding tests were carried out without preheating (Table 1). Chemical composition and mechanical properties of the S700MC steel are presented in Table 1 [45]. The gap between sheets was varied in the range 0–3 mm. The best results were obtained for gap = 1.5 mm. This case was taken for further tests only.

**Table 1.** Welding parameters of S700MC steel.

| Layers Order | Welding Method | Diameter of the Electrode, mm | Current Intensity, A | Voltage, V | Polarization | Welding Speed, m/min | Input Energy, kJ/cm |
|---|---|---|---|---|---|---|---|
| 1 | 135 | 1.0 | 100 | 19 | DC "+" | 300 | below 4 |

Micro-jet cooling parameters were slowly varied:

- Micro-jet gas: argon
- Stream diameter: 60 and 70 μm
- Gas pressure: 0.6 and 0.7 MPa.

Once all of the test were performed the following quality control checks were applied: non-destructive and destructive tests.

Non-destructive tests included (NDT):

- Visual testing (VT) of the manufactured welded joints was performed with an eye armed with a loupe (Levenhook, Tampa, FL, USA) at $3\times$ magnification—tests were carried out in accordance with the requirements of the PN-EN ISO 17638 standard, evaluation criteria according to the EN ISO 5817 standard.

Visual testing of welds was made using standard auxiliary measures, luxmeter with white light 520 Lx. It was found that only some of the tested welds were made correctly and met the quality requirements; they were characterized by the limit of acceptability "B" according to PN-EN ISO 5817 [43]. Magnetic-particle test of welds was made using the wet method with the following conditions: field strength 3 kA/m, white light 515 Lx, temperature 20 °C, MR-76 detection means, MR-72 contrast.

- Magnetic-particle testing (MT)—the tests were carried out in accordance with the PN-EN ISO 17638 standard, the evaluation of the tests was carried out in accordance with the EN ISO 5817 standard, the device for testing was a magnetic flaw detector of REM—230 type (ATG, Prague, Czech Republic).

The destructive tests included:

- Visual tests on micro-sections of welded joints were performed with an eye armed with a loupe at $3\times$ magnification—tests were performed according to PN-EN ISO 17638 standard with reagents for testing according to PN-CR 12361 standard, evaluation criteria according to EN ISO 5817 standard;
- The bending test was carried out in accordance with the PN-EN ISO 5173 standard, using the ZD-40 testing machine (WPM, Leipzig, Germany).

Then, a bending test was performed only for those specimens that passed the NDT test. Thus bending test was carried out only for specimens without cracks (with B acceptability). The tests used: specimens with a thickness of a = 3 mm, width b = 20 mm, mandrel d = 22 mm, spacing of supports d + 3a = 31 mm, and the required angle of bending 180. Five bending test measurements were carried out for each tested joint thickness on the root side and on the face side. Specimens welded with both electrode wires (UNION X90 and UNION X96) with two variants (with micro-jet cooling and without micro-jet cooling) were selected for the tests: with (a) argon micro-jet cooling with stream diameter 60 μm and gas pressure 0.7 MPa, wire UNION X90, (b) with argon micro-jet cooling with stream diameter 60 μm and gas pressure 0.7 MPa, wire UNION X96, (c) without micro-jet cooling, wire UNION X90, (d) without micro-jet cooling, wire UNION X96.

- Examination of microstructure of specimens etched with Adler reagent using light microscopy (Neophot 32, Carl Zeiss Jena, Jena, Germany);

Samples prepared for observation after welding with electrode wire UNION X90 with two variants (with micro-jet cooling and without micro-jet cooling). The area of the parameters of the preferred micro-jet cooling was narrowed down again for the analysis of the M90-60-07 sample and the W90 sample that corresponded with: (a) MAG welding with argon micro-jet cooling with stream diameter 60 μm and gas pressure 0.7 MPa, wire UNION X90, (b) MAG welding without micro-jet cooling, wire UNION X90.

- Tensile and fatigue tests.

Tensile and fatigue tests were conducted using the 8874 INSTRON servo-hydraulic testing machine (Instron, High Wycombe, UK) and mini-specimens at room temperature. The following standards PN-EN ISO 6892-1:2020 [46] and ASTM E468-18 [47] were used, respectively. The specimens were directly selected from parent material and region with weld, manufacturing flat (Figure 2a) and hourglass specimens (Figure 2b) for monotonic and cyclic tension tests, respectively. Tensile experiments were carried out by means of three types of loading signal i.e., displacement, strain, and stress, which have the following values of velocity: 1 mm/min, 0.08 1/min, and 178 MPa/min, respectively. The axial strain of specimens subjected to tension was captured by means of an axial extensometer on gauge length of 12.5 mm (Figure 2a). The specimens for fatigue tests were designed basing on the requirements of the following standards: ASTM E466-15 [48] and ASTM E468-18 [47]. The

face and root of the mini-specimens were removed in technological process with respect to examining the weld at a defined stress state. Worth to notice that results on the specimens with a smooth measuring section can be easy compared with the data on parent material. Moreover, the removal of the face and root of the weld allows the quality of the joint to be assessed directly in the zone of the used parts. Nevertheless, from the engineering point of view, both kinds of specimens should be used because results of modelling and technical data as well as elaborating stages of inspections can be covered easier. In the proposed experimental procedure only specimens with the smooth measurement section were employed with respect to this feature which appears in a lot of specimens used in experimental mechanics of materials and its degradation can be followed at a small number of technical limits. Moreover, the measuring region in the form of an hourglass has directly enabled testing the weld under selected loading conditions. The weld zone was of the physical fracture plane for damages initiation, coalescence, and growing up to fracture. This kind of specimen is also much less sensitive to any scratches and big roughness because of the narrow fracture region. Therefore, this approach was used to shape and prepare the specimens. Worth to emphasis that from a practical point of view, experiments on welded specimens having a face and root can follow the additional piece of the knowledge because all sections of a joint can be tested. Comparing data from experiments on machined [49] and non-machined specimens [50], differences in weld behavior can be evidenced [51,52]. Nevertheless, this stage will be considered in the next paper, as an additional approach to examining the micro-jet cooling welded joints of S700MC steel.

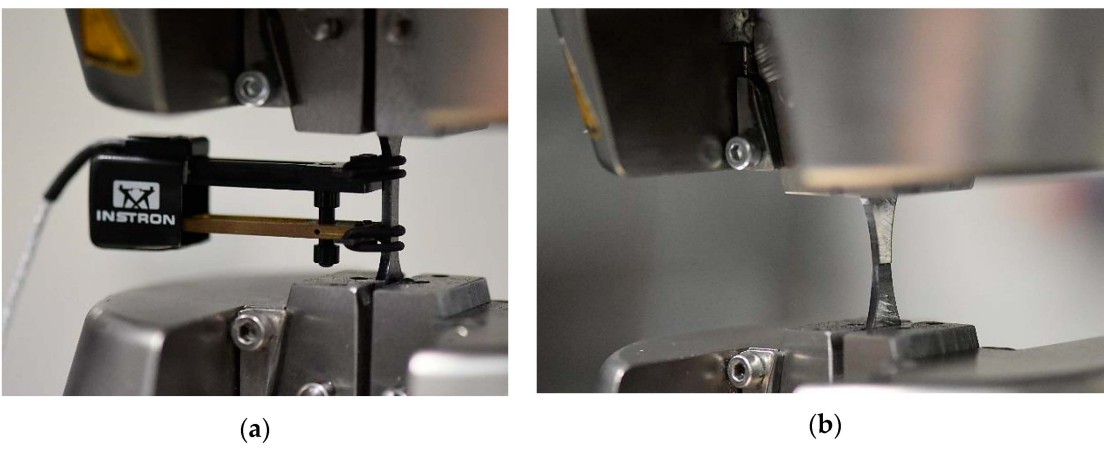

| (a) | (b) |

**Figure 2.** Flat (**a**) and hourglass (**b**) mini-specimen in testing machine before tensile and fatigue test, respectively, nominal dimensions of the cross sections in measuring zones: 4 mm × 2 mm (**a**); 4 mm × 3 mm (**b**).

In the case of fatigue test, with respect to dynamic fracturing of the weld tested, which would lead to failure of the extensometer, a displacement collected by the sensor of the testing machine was used for assessing the weld deformation. All tests were performed employing stress signal at a maximum value between of 400 MPa and 950 MPa at stress ratio R = $\sigma_{min}/\sigma_{max}$ = 0.1 and frequency 10 Hz. During fatigue examination, values of the following physical quantities were recorded: force (Figure 3(a1)), stress (Figure 3(b1)), displacement (Figure 3(a2,b2)), total energy (Figure 4), time, and cycles. Data in a form of displacement versus a number of cycles were taken to assess the weld behavior under cyclic loading, expressed by cyclic hardening and softening at initial and subsequent cycles, respectively (Figure 3(a2,b2)). They were collected in various types of files, which follows: courses of the physical quantities, their maximal and minimal values, and values directly before and during fracture. In the case of the last stage, the 50th last cycles before decohesion were used for determining the weld behavior at the final stage of the material tested. With respect to extending the knowledge on the weld behavior under stress levels, fracture regions were analyzed using the macro-photography technique.

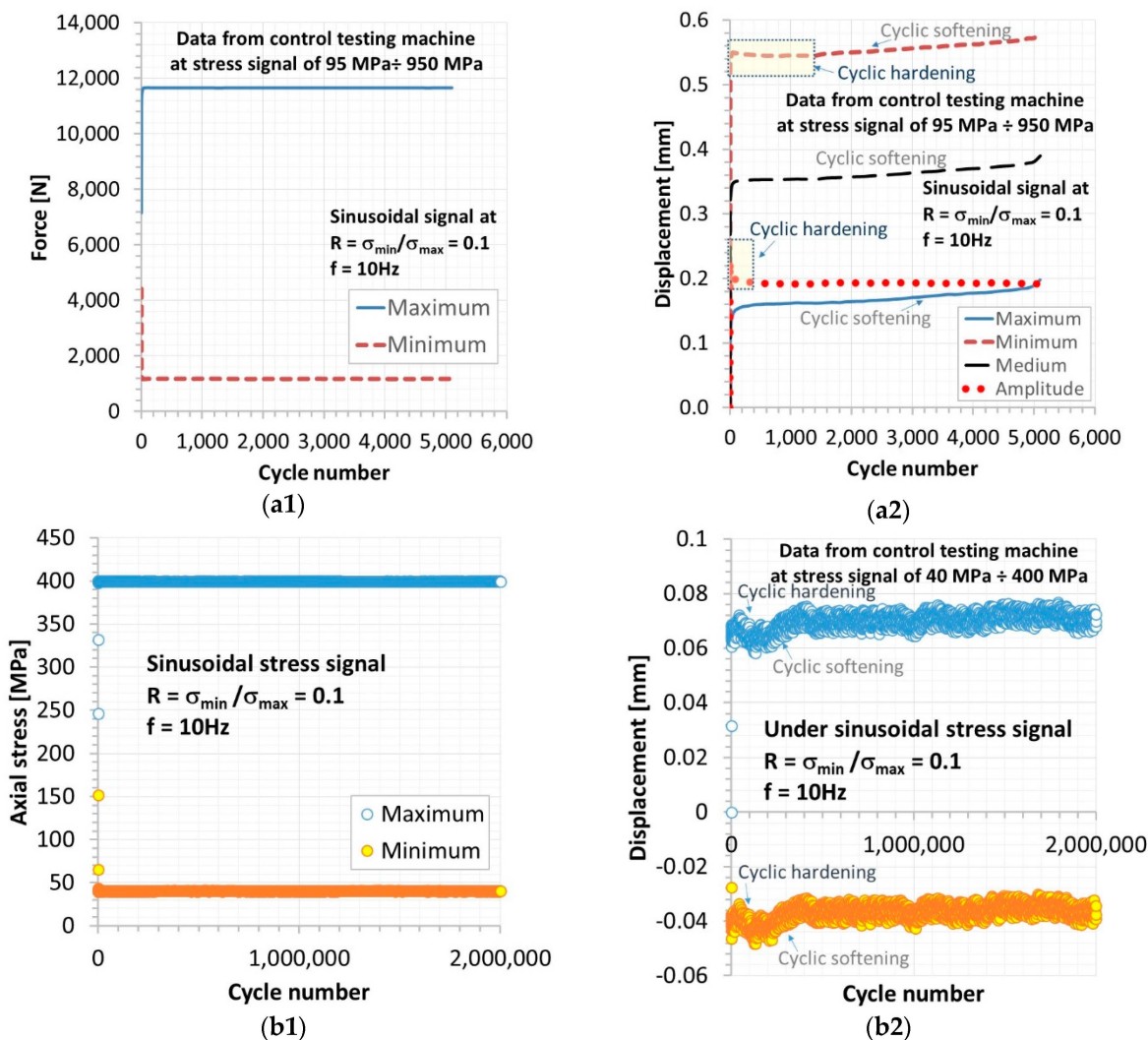

**Figure 3.** Maximum and minimum values of force (**a1**) and stress signal (**b1**) and displacement (**a2,b2**) from the fatigue test for determining fatigue limit of S700MC steel welded by means of the welding wire of Union X96, reaching $5 \times 10^3$ (**a1,a2**) and $2 \times 10^6$ cycles (**b1,b2**).

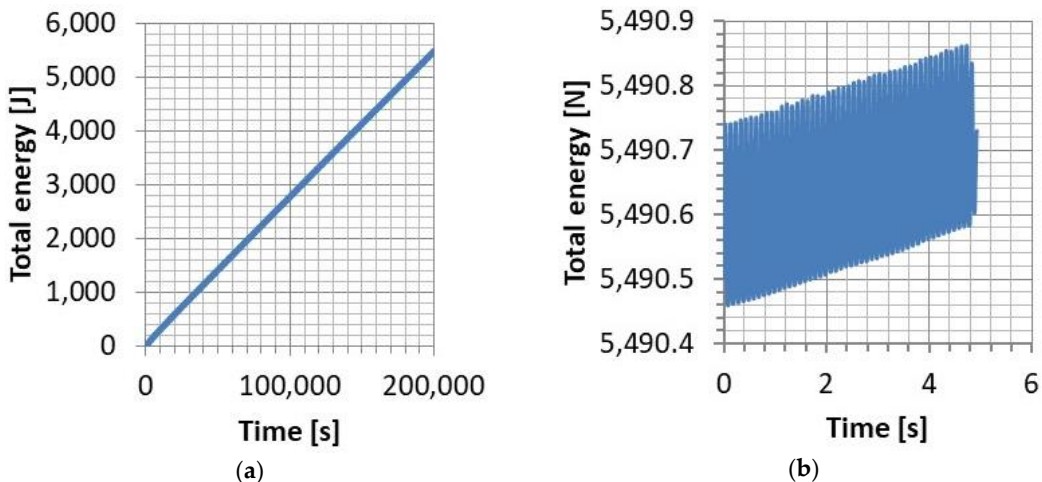

**Figure 4.** Courses of total energy collected in fatigue test up to fatigue limit (at $2 \times 10^6$ cycles) of S700MC steel welded using the Union X96 wire: (**a**) from the beginning of the test up to the stage directly before of the limited number of cycles, (**b**) the last 5 seconds at the final stage of the test up to the moment determining fatigue limit i.e., 400 MPa.

## 3. Results

### 3.1. The Results of Non-Destructive Tests

Non-destructive test results are presented in Table 2. They expressed differences in the weld quality and confirmed a meaning of this stage concerning assessment of the quality of zone examined.

**Table 2.** Assessment of non-destructive testing of the movable platform joint.

| Sample Designation | Micro-Jet Stream Pressure MPa | Micro-Jet Stream Diameter μm | Wire | Micro-jet Gas | Observation, Acceptability |
|---|---|---|---|---|---|
| W90 | without | without | Union 90 | without | Cracks in the weld |
| M90-60-06 | 60 | 0.6 | Union 90 | Ar | Cracks in the weld |
| M90-60-07 | 60 | 0.7 | Union 90 | Ar | No cracks, B |
| M90-70-06 | 70 | 0.6 | Union 90 | Ar | No cracks, B |
| M90-70-07 | 70 | 0.7 | Union 90 | Ar | Cracks in the weld |
| W96 | without | without | Union 96 | without | Cracks in the weld |
| M96-60-06 | 60 | 0.6 | Union 96 | Ar | Cracks in the weld |
| M96-60-07 | 60 | 0.7 | Union 96 | Ar | No cracks, B |
| M96-70-06 | 70 | 0.6 | Union 96 | Ar | No cracks, B |
| M96-60-06 | 70 | 0.7 | Union 96 | Ar | Cracks in the weld |

### 3.2. The Bending Tests

The test results are summarized in Table 3. From the analysis of the results presented in Table 3, it follows that the test was carried out correctly, the evaluation of the tests is positive only in some cases when micro-jet cooling was not too intense and not too weak. Process was correct only with medium power micro-jet cooling, because no cracks and other disconformities were found in the samples tested. By changing the welding parameters (current, welding speed), it is possible to avoid cracks, but it is not possible to obtain high plastic properties of welds with the relative elongation at the level of base material properties. This was examined in microstructural analysis for the weld tested as well as fracture zones after fatigue tests by means of micro-photographic techniques. These approaches did not express any cracks in the weld, which is indicated by the pure quality of the region manufactured. It reflected the high-quality level of the joint and it was confirmed in comparison to the regimes of the PN-EN ISO 5817 standard [52] on a crack length $\geq 0.5$ mm at the highest requirement denoted by letter B.

**Table 3.** Bending tests results.

| Sample Designation | Deformed Side | $a_o$ (mm) $\times$ $b_o$ (mm) | Bending Angle (°) | Notes |
|---|---|---|---|---|
| M90-60-07 | root of weld | 3.0 × 20.0 | 180 | no cracks |
| M90-60-07 | face of weld | 3.0 × 20.0 | 180 | no cracks |
| M96-60-07 | root of weld | 3.0 × 20.0 | 180 | no cracks |
| M96-60-07 | face of weld | 3.0 × 20.0 | 180 | no cracks |
| W90 | root of weld | 3.0 × 20.0 | 180 | cracks |
| W90 | face of weld | 3.0 × 20.0 | 180 | cracks |
| W96 | root of weld | 3.0 × 20.0 | 180 | cracks |
| W96 | face of weld | 3.0 × 20.0 | 180 | cracks |

### 3.3. Metallographic Examination

Observations of the samples digested in Adler's reagent were carried out on the Reichert light microscope. The examined joints are dominated by a martensitic and ferritic structure—Figure 5 shows the structure of the W90 sample.

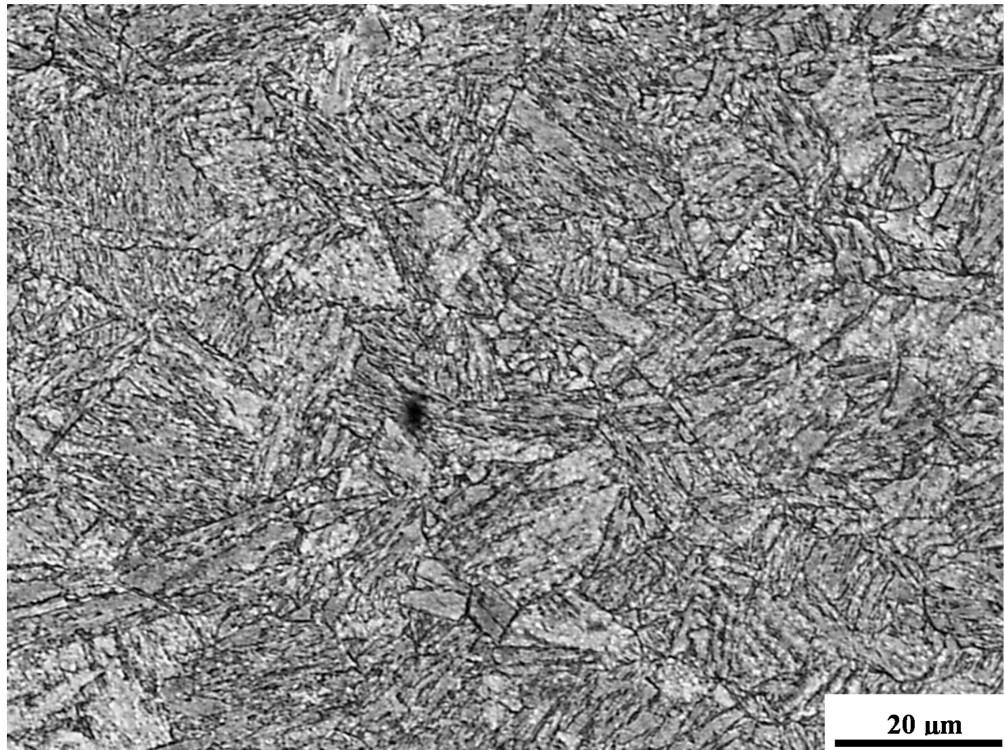

**Figure 5.** Microstructure of the joint (W90). Visible martensite and course ferrite (LM).

In addition to martensite, coarse-grained ferrite was observed. Only the application of micro-jet cooling after MAG welding made it possible to obtain much more fragmented ferrite (shown in Figure 6).

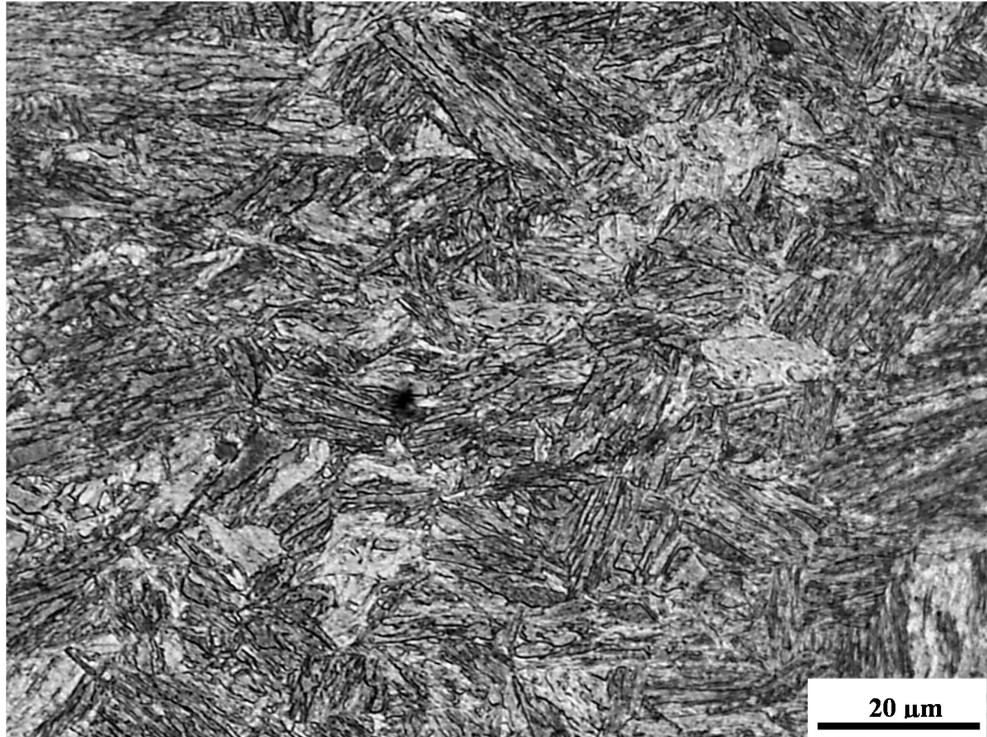

**Figure 6.** Microstructure of the joint (M90-60-07) after welding with micro-jet cooling. Visible martensite and fine-grained ferrite (LM).

Looking at Figures 5 and 6, it can be concluded that the structure appears to be similar. In Figure 6, it is visible that the ferrite is slightly more fragmented. This is due to the use of micro-jet cooling during welding, which affects the grain growth during the austenitic transformation. Other authors also draw attention to the possibility of grain grinding during welding of steel with the use of micro-jet cooling [17,19].

An important role of microstructure in the steel behavior under loading can be evidenced in the tensile characteristic, which expressed values of yield stress and ultimate tensile strength above 800 MPa at attractive ductility taking of 17.5%, indicating beneficial features of the steel tested in comparison to other types of this kind of material (Figure 7a). This kind of microstructure also results in the stability of the stress parameters (Figure 7b) and similarity of fracture zones (Figure 8) at various types of loading signal, containing: displacement, strain, and stress. It means that in the case of the weld, this type of microstructure is very desired and its manufacturing extends the application range of components with this kind of welded regions, containing different types of loading.

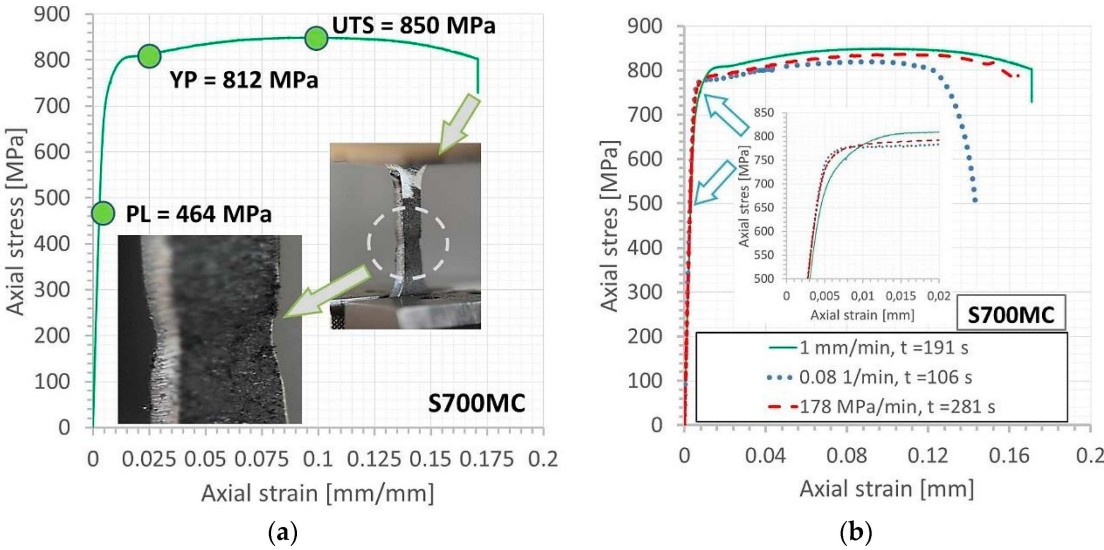

(**a**)                                              (**b**)

**Figure 7.** Tensile characteristics of S700MC steel from tests conducted at various loading signals at the following values of velocity for: (**a**) displacement 1 mm/min, (**b**) strain—0.08 1/min.

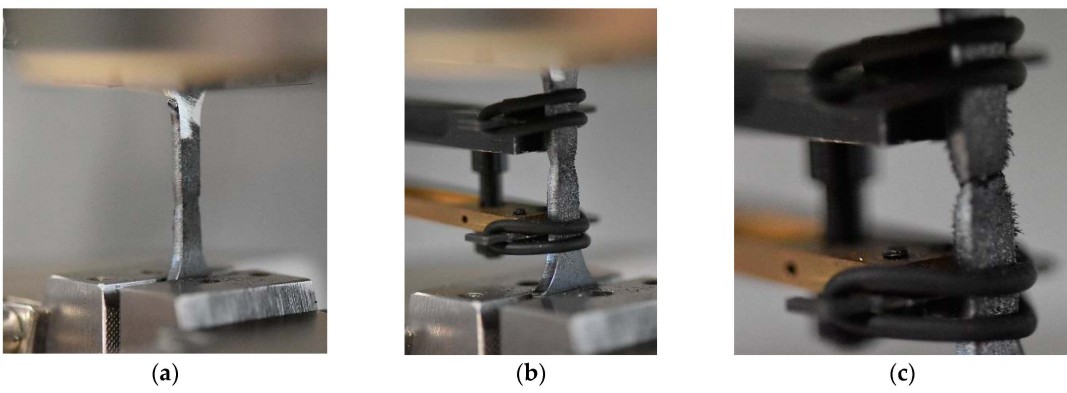

(**a**)                              (**b**)                              (**c**)

**Figure 8.** S700MC steel at final stage of tensile tests conducted at the following values of velocity of loading signal: (**a**) 1 mm/min (displacement), (**b**) 0.08—1/min (strain), (**c**) 178 MPa/min (stress).

As it was determined in the tensile tests conducted on the S700MC steel joint produced by the use of the welding wire named the Union NiMoCr, EDFK 1000 and Union X96, that all results exhibited very similar values of ultimate tensile strength and they were only 5% lower than the value of this parameter for the parent material (Figure 9a). Differences in

elasto-plastic region up to the necking point are more than evident. Nevertheless, fracture zones expressed variations in the mechanism of degradation from shear fracture (at the welding wires of Union NiMoCr and EDFK 1000) to ductile fracture (for the Union 96 welding wire) (Figure 10a,b, respectively). In comparison with the features of the fracture region of the parent material after tension (Figure 8) the ductile degradation is dominant in the region examined. A significant improvement in plastic properties was demonstrated in relation to the achievements of previous researchers. The relative elongation of the joint was achieved at the level of the parent material, which other authors did not succeed. Therefore, the weld has been selected for tests under cyclic loading (Figures 2b, 3, 4, 11–14). Nobody has analyzed the welding of S700MC steel using an innovative process using micro-jet cooling so far, and nobody has checked the fatigue strength of such welds. Results captured from this kind of tests are represented by fracture regions (Figure 11), variations of displacement (Figure 12), and total energy (Figure 13) at final cycles as well as the Wöhler curve (Figure 14).

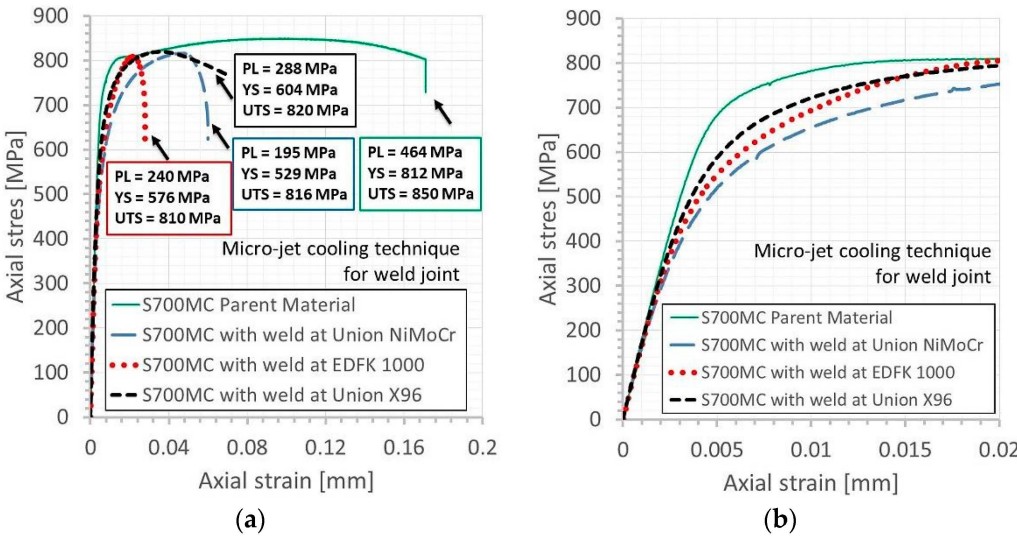

**Figure 9.** Tensile characteristics of S700MC welded by means of three different welding wires in: (**a**) general view, (**b**) focuses on elastic and elastic-plastic regions.

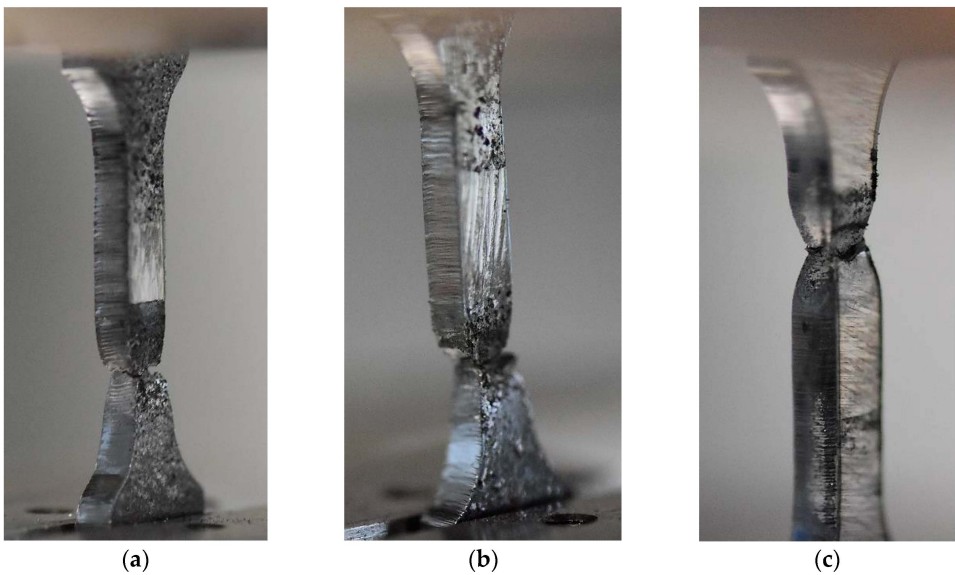

**Figure 10.** Fracture zones of tensile specimens with joints manufactured at the welding wires: (**a**) Union NiMoCr, (**b**) EDFK 1000, (**c**) Union X96.

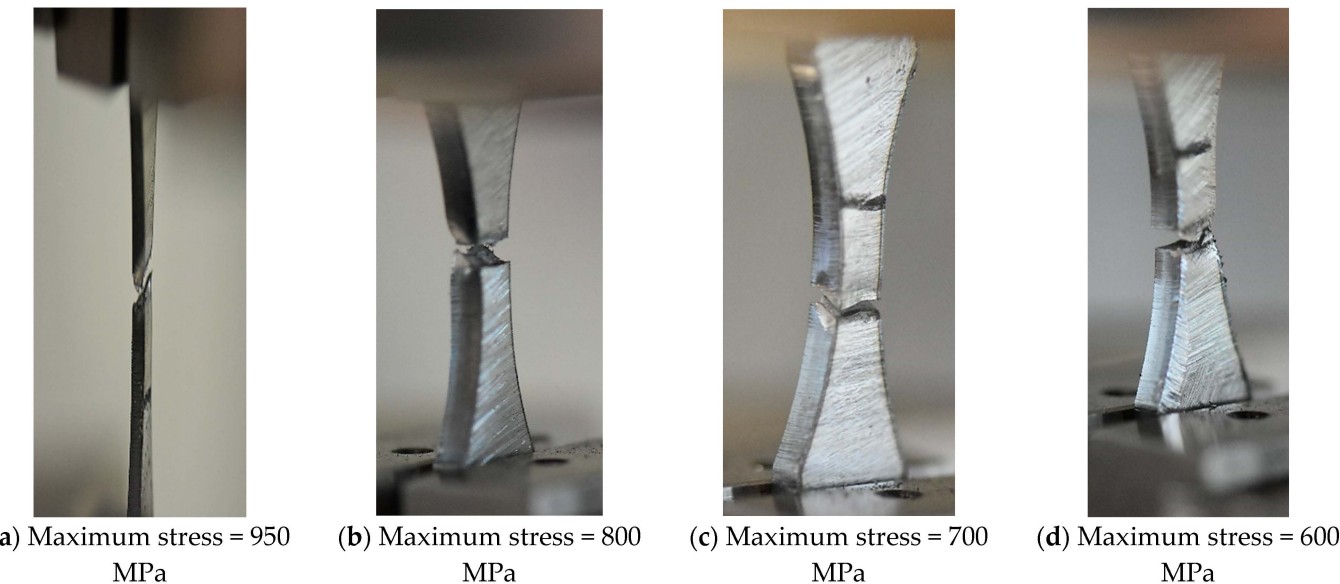

(**a**) Maximum stress = 950 MPa　　(**b**) Maximum stress = 800 MPa　　(**c**) Maximum stress = 700 MPa　　(**d**) Maximum stress = 600 MPa

**Figure 11.** Fracture zones of S700MC steel welded at supporting the micro-jet cooling technique at various values of maximal stress after fatigue test, i.e. (**a**) 950 MPa, (**b**) 800 MPa, (**c**) 700 MPa and (**d**) 600 MPa.

Data captured from fatigue tests have enabled to follow difference in the degradation of the weld examined at various stress levels (Figure 11). They were represented by reorientation of fracture zone from angular (Figure 11a–c) to horizontal (Figure 11d) at a high and low value of stress applied, respectively. Connecting these results with tensile mechanical properties of the weld (Figure 9) the stress value related to the change of degradation mechanism can be indicated. In this case, the yield stress plays this role being a major factor in the weld behavior under the stress range considered.

Some differences in the weld response on the stress values applied are collected based on data in a form of displacement versus time (Figure 12), but they were not significant as taken from the fracture regions. They show a limit value of displacement before the appearance of weld fracture, i.e., 3 mm, indicating on the almost same level of deformation at crack growing. This occurs under unloading, resulting in a rapid reveal of the fracture zone. Nevertheless, the final stage connected with decohesion appeared later and at different values of deformation recorded and it is strongly related to the maximum value of stress used (Figure 12a–c). This also is represented by no differences between the stages following deformation increasing and decohesion at stress value below the yield point (Figure 12d). It can be explained based on the hardening of the weld at higher values of stress and its small scale at lower ones. Moreover, following these data (Figure 12a) and results from tensile test (Figure 7) the influence of the hourglass measuring zone on the weld response can be evidenced as the stress value exceeding the ultimate tensile strength.

The similarity to conclusions from fracture zones (Figure 11) and displacement courses (Figure 12) total energy versus time (Figure 13) enables to connect its values with stress levels used. Besides, in the case of a high value of stress this parameter increased more rapidly than for the smallest ones, a final value of total energy, representing its huge increase was dependent on the stress levels, taking 5 J (calculated as of 1976 J–1981 J) at 950 MPa (Figure 13a) and two times smaller value at stress, not exceeding the yield point (Figure 13c,d). It can be explained in the same way as in the case of displacement courses i.e., by weld hardening.

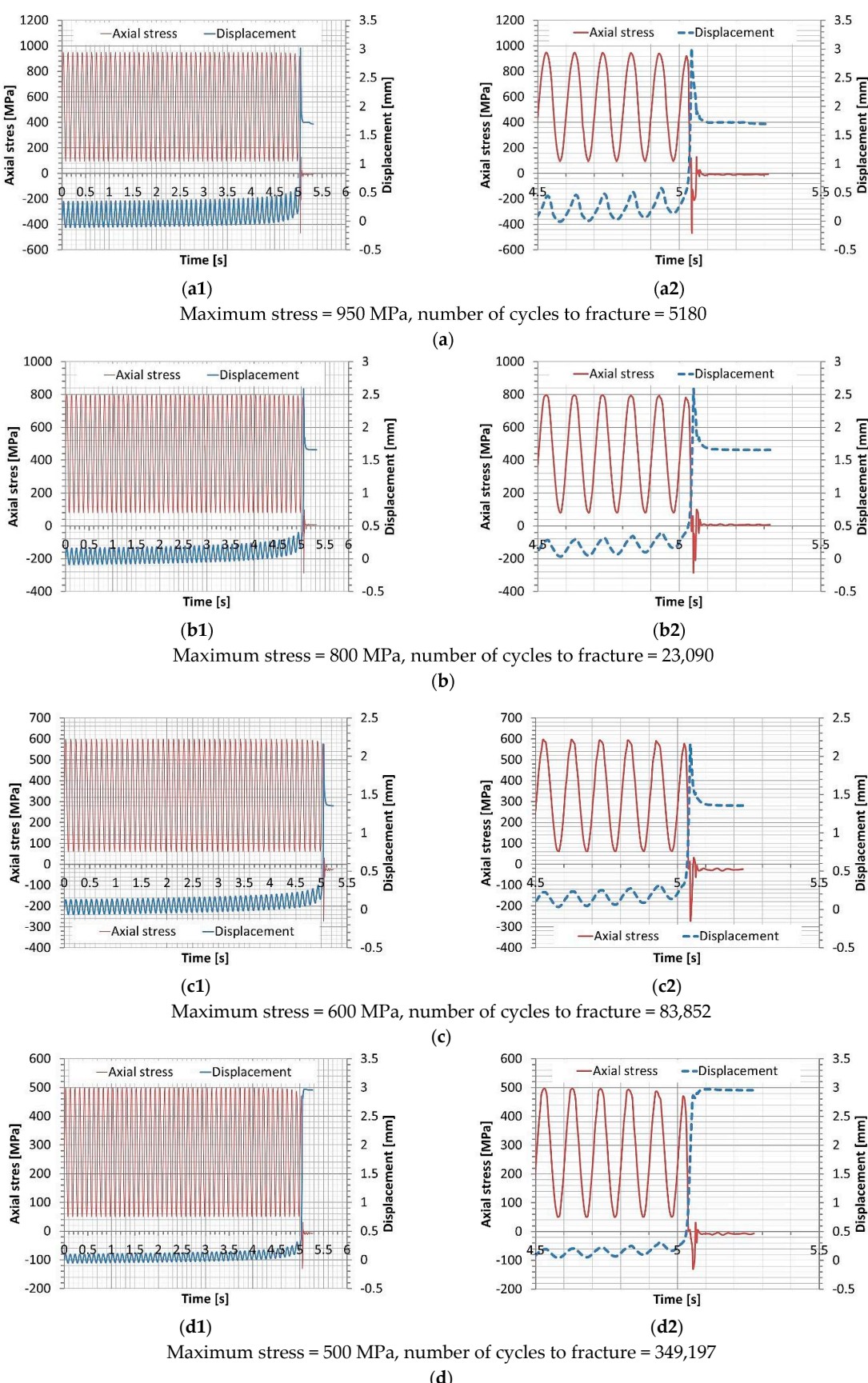

**Figure 12.** Variations of axial stress and displacement before fracture of the weld under the following values of maximum stress: (**a**) 950 MPa, (**b**) 800 MPa, (**c**) 600 MPa, (**d**) 500 MPa.

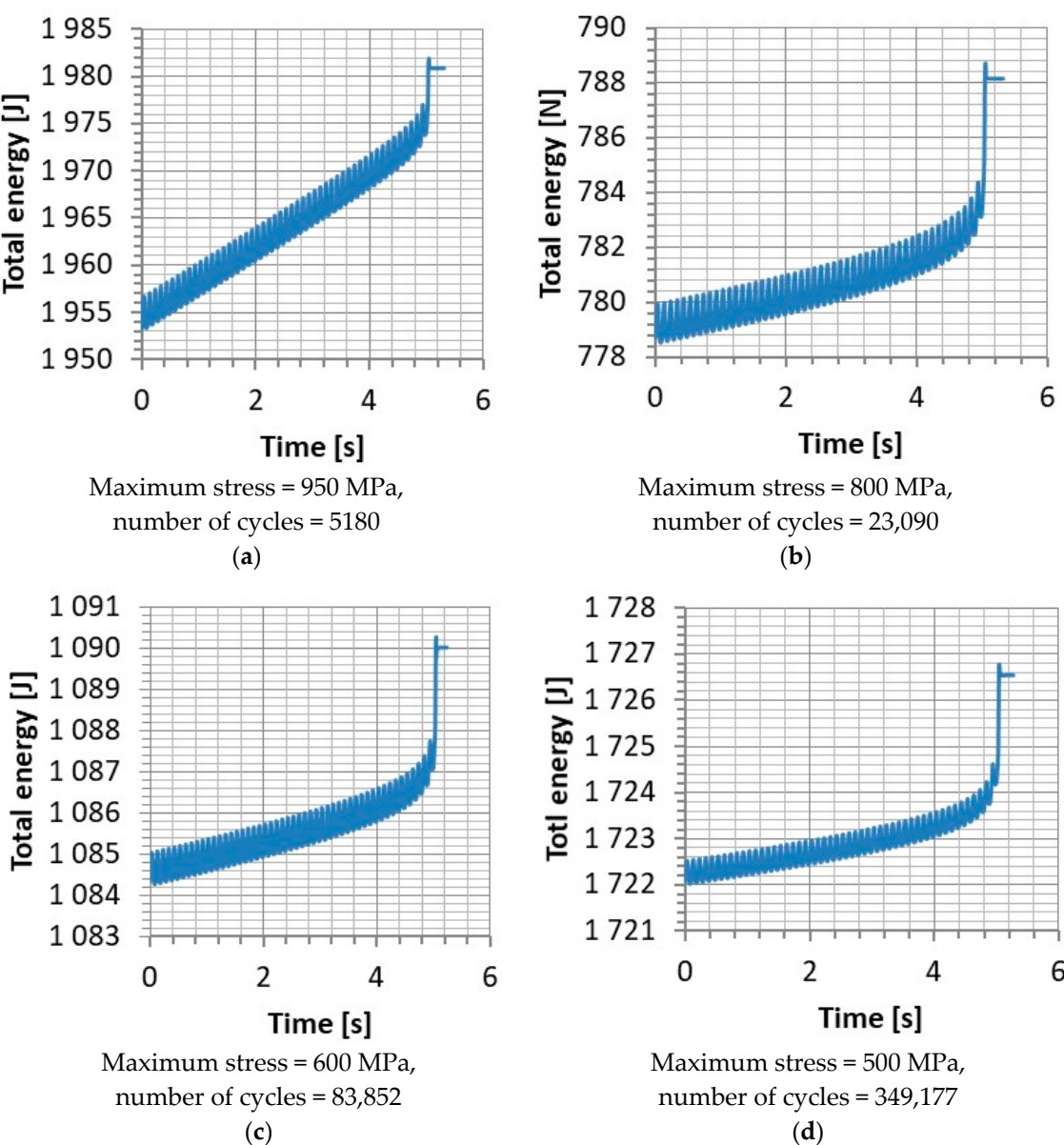

**Figure 13.** Total energy versus time before fracture of the weld under the following values of maximum stress: (**a**) 950 MPa, (**b**) 800 MPa, (**c**) 600 MPa, (**d**) 500 MPa.

Generally, differences in the weld behavior under stress level applied were clearly noticed on the relation between axial stress and number of cycles (Figure 14). They were represented by time to fracture containing a few cycles at the highest stress applied 1400 MPa and $2 \times 10^6$ for the smallest one i.e., 400 MPa which was established as a fatigue limit of the welded joint because fracture did not occur. Analyzing these data and results from the tensile test, a number of cycles under stress value related to fundamental mechanical properties can be indicated, giving a piece of the knowledge for engineering and researches groups, which use software for designing, calculations, and modelling. They can be directly used for validation of the statistical approach and for elaboration with respect to increasing the prediction accuracy at small-number specimen life data [53].

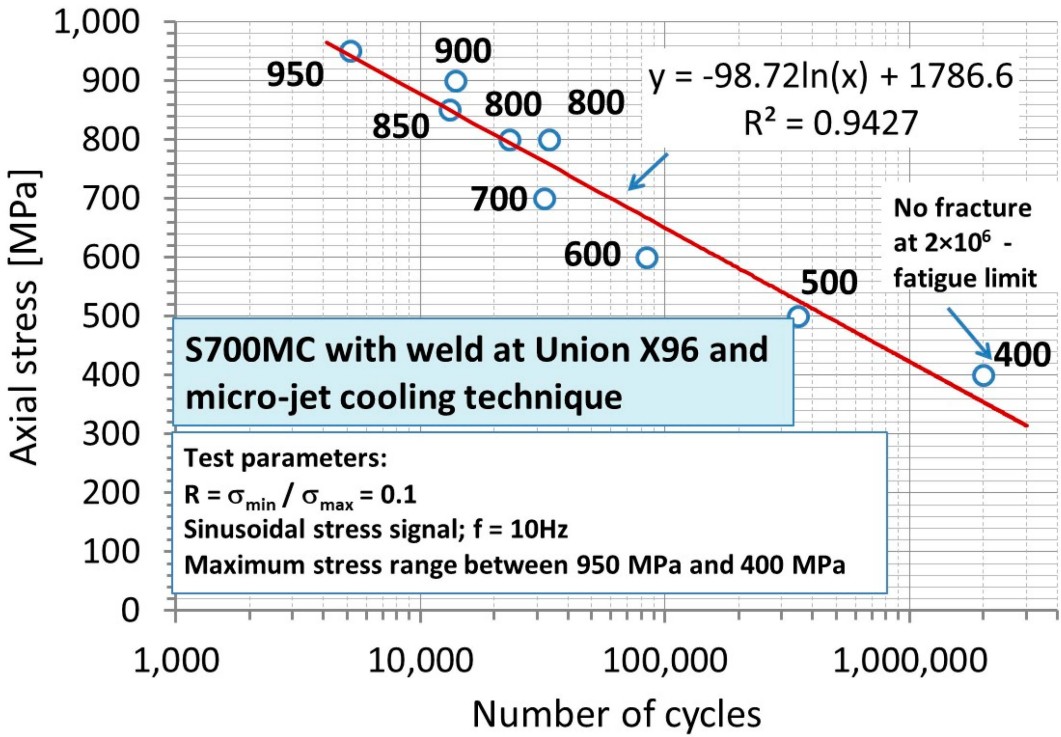

**Figure 14.** Wöhler curve of S700MC steel welded by means of the welding wire of Union X96.

## 4. Summary

The article deals with welding of S700MC steel with the use of micro-jet cooling. Welds were made with different electrode wires, various welding parameters and various parameters of micro-jet cooling. After the results of non-destructive tests, main information was obtained on the proper welding parameters, where no cracks appear. Then, the temporary tensile tests were performed and the fatigue experiments were carried out. An influence of the welding process with micro-jet cooling on the steel behavior was represented by 50% reduction of elongation at an almost constant value of ultimate tensile strength. The metallographic structure of the joint was checked. The Wöhler curve was determined and the fatigue limit was presented. These results have enabled to indicate the value of fatigue limit of the micro-jet cooling high strength weld i.e., 400 MPa and its relationship to ultimate tensile strength denoted by 0.48 which supports the efforts of engineers and researches on the elaboration of the welding process and modelling as well as inspection stages. The weld response at initial fatigue cycles exhibited the hardening, which directly led to an increase in the stress value used to control the testing machine. This effect disappeared with an increasing number of cycles changing to softening of the weld tested. It was found that, owing to micro-jet cooling, very good mechanical properties of the joint were obtained, as measured by the excellent results of fatigue tests.

## 5. Conclusions

Micro-jet cooling has been successfully used to weld S700MC steel. It was the first time such a joint was made and its mechanical properties were thoroughly tested. A novelty is also checking the fatigue properties for joints made with using micro-jet cooling.

Based on the research carried out, the following conclusions can be drawn:

1. Good mechanical properties can be obtained when welding S700MC steel by using micro-jet cooling.
2. The properties of the S700MC joint are affected by the selection of thermodynamic welding conditions, including the micro-jet cooling parameters.

3. The conducted non-destructive and destructive tests confirm the correctness of using micro-jet cooling during welding of S700MC steel.
4. The result of the fatigue tests allows us to state that the proposed welding process can be applied to responsible structures.

**Author Contributions:** Conceptualization, T.W., T.S. and B.S.-L.; methodology, B.S.-L.; software, T.S.; validation, T.W., T.S. and B.S.-L.; formal analysis, B.Ł.; investigation, T.S.; resources, B.S.-L.; data curation, T.W.; writing—original draft preparation, T.S.; writing—review and editing, B.S.-L.; visualization, T.S.; supervision, B.Ł.; project administration, T.W.; funding acquisition, B.S.-L. All authors have read and agreed to the published version of the manuscript.

**Funding:** This research was funded by (Silesian University of Technology) grant number (BK-205/RT1/2020) and the APC was funded by (Metals (voucher)).

**Institutional Review Board Statement:** Not applicable.

**Informed Consent Statement:** Not applicable.

**Data Availability Statement:** Not applicable.

**Acknowledgments:** The paper is part of the COST project, CA 18223.

**Conflicts of Interest:** The authors declare no conflict of interest.

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
