# Peer review of "MAG Welding Process with Micro-Jet Cooling as the Effective Method for Manufacturing Joints for S700MC Steel"

_metals, doi:10.3390/met11020276_

Round 1
Reviewer 1 Report
Scientific paper about weldability and fatigue performance of AHSS steel S700MC. Authors conclude on the benefit of using micro-jet cooling, but this technique / system is not explained or shown in the paper with minimum detail. Better description and images of this system must be included.
The structure of the paper must be improved since experimental methods, results and conclusions are mixed in section 3. Some experimental details are duplicated in section 2 and 3, whereas others are missing.
I was not able to understand whether two or three different filler metals were employed in the study.
A long list of standards followed during the study was given, but the most important ones (tensile and fatigue standards) are not mentioned.
One only can clearly conclude that MJC avoids cracks under particular conditions. Maximum crack length acceptance level for EN ISO 5817 class B must be indicated in the paper.
It is not clear to me how fatigue tests were carried out at 950 MPa maximum stress or even up to 1400 MPa in the S-N curve, if the UTS of either parent or welded samples was lower than 850 MPa.
Line 17 - Consider revision of this sentence "New technology, micro-jet cooling (MJC) after welding with micro-jet cooling could be the"
Line 21 - Analysis
Line 47-48 - S700MC together
Line 58 - include what CET stands for
Line 59 - New paragraphs starting at "In this paper".
Line 84 - Flat psoition (PA)
Lin 90 - include a table for filler metals and chemical composition. Keep the same filler metal traceability in the whole paper.
Line 97 - less splinters and machining??
TAble 1 - erase last row
Lines 109-110 - avoid duplications
Line 134 - digested?? I suggest etched
Line 136 - Indicate standards followed for tensile and fatigue tests
Line 142 - 178 MPa/min
Line 162 - avoid duplications
Line 182 -186 - Tis info is already given in TAble 5.
Line 195. What is Union X9??
Line 202 - Figure 5 is for W90, accroding to its caption.
Line 208 - Micros in Fig. 5 and Fig.6 seem comparable. A better explanation and indication of ferrite fragmentation must be given.
Lines 218-219 - In comparison with other ones??? I cannot understand meaning of this statement.
Line 226 - Reference of filler metals???
Line 228 - Difference in elasto-plastic region up to the necking point are more than evident.
Lines 233-234 - Not clear what "this kind of weld" means.
Line 257-258 - explain where 5J total energy values was taken from.
Line 270 - 4. Conclusions
Author Response
Thank you for you kind review. Happy New Year 2021

Reviewer 2 Report
- The title is too long, please shorten it.
- The conclusion or summary is like a process description and no useful information inside. Please provide a real conclusion and also a quantitative point of you investigation.
- no data is found in the regime of million cycles, please also add more data points in Fig. 14. Also how do you derive the fatigue limit from this picture. With regard to producing PSN curve, please add one paper from "10.1016/j.ijfatigue.2020.105789".
- Fig. 10 and 11 indicate a big roughness and clear scratch of the specimen and such surface quality can influence the fatigue lifetime of joints particularly for high cycle fatigue case, you should give more details about this experiment and process.
- Fig. 5 and 6 should clearly denote the location at the joint using SEM for local microstrucures. Also the picure has a poor quality, please improve it. EBSD is better to replace current pictures.
- This paper should clearly denote its creativity or originality since this work always cite the reference [13].
- Figure 1 should be simplified into one picture and give the thickness of the plate.
- Finally, the paper must be reorganized clearly.
Author Response

(The authors gave the same response as above.)

Reviewer 3 Report
Sorry, but there some severe concerns about this paper:
"relatively not very well weldable" what does that mean?
Introduction with smart city 1.0 is irritating/misleading
"thin sheet steel with a thickness range from 0.4 mm to 16 mm" That is wrong! thin sheets is only up to 3.0 mm by definition!
Introduction is not focussed: AHSS, automotive, 16 mm, hydrogen...
fatigue strength of steels in welded components/Automotive is no issue, only the question of design!!! See also Eurocode 3!
what is the targeted application of S700 in 3 mm???
Fatigue strength in constructions has to be tested with relevant seam profiles. Flat specimens (weld reinforcement removed) are purely academic!
Why do you cite 38 references when you do not use a single one to discuss your results??? This is NO scientific paper!!!
Author Response

(The authors gave the same response as above.)

Reviewer 4 Report
S700MC steel, as a low carbon steel, generally has good weldability. Various researchers have focused on the performance of the welded S700MC steel. Though the researchers presented some experimental results in the article, the analysis is in lack. The manuscript is more next to a report, rather than a scientific paper.
Though the authors reported that microjet cooling reduced the cracks. The reviewer is curious that can we avoid the cracks by adjusting the welding parameters, e.g., current, welding speed, and so on?
What is the suggested weldable parameter window?
Generally, reducing crack sensitivity by cooling is a widely used method. The innovation of this manuscript is at the common level.
The figures are not well prepared.
In the introduction section, much more literatures about the knowledge gap should be clearly identified.
Author Response

(The authors gave the same response as above.)

Round 2
Reviewer 1 Report
Thank you for your detailed answers.
I was not able to find the corrected version of the manuscript, so I guess that you introduced your corrections in the updated version.
I still do not understand how fatigue samples tested at axial stress amplitude of 950 MPA (R=0.1, Rmax 950 MPa and Rmin 95 MPa) can withstand 5000 cycles if the UTS of this steel was 850 MPa and even lower than 820 MPa for the welded samples.
Round 2
Dear Reviewer
Thank you very much for your second efforts for improving the paper. In our opinion you suggestions make the paper follows the theme in a more correct manner than it was presented at the beginning. We appreciate your support and we would like to express our thank you again.

Reviewer 2 Report
I have no further questions.
Author Response
Dear Reviewer
Thank you very much for your efforts for improving the paper. In our opinion you suggestions make the paper follows the theme in a more correct manner than it was presented at the beginning. We appreciate your support and we would like to express our thank you again. Authors
Reviewer 3 Report
Was not really improved! the main point of useless fatigue testing without weld reinforcement was not changed.
Discussion is still not enough!!! Only a fraction of the cited literature was used....
Author Response
Dear Reviewer
Thank you very much for your efforts for improving the paper. In our opinion you suggestions make the paper follows the theme in a more correct manner than it was presented at the beginning. We appreciate your support and we would like to express our thank you again.

Reviewer 4 Report
The authors answered some of the questions except for the following one.
What is the suggested weldable parameter window that could produce the joint with good performance?
Author Response
Dear Reviewer
Thank you very much for your efforts for improving the paper. In our opinion you suggestions make the paper follows the theme in a more correct manner than it was presented at the beginning. We appreciate your support and we would like to express our thank you again,
Authors
